# The Ground-Level Particulate Matter Concentration Estimation Based on the New Generation of FengYun Geostationary Meteorological Satellite

Lin Tian [1,2], Lin Chen [1,2,*], Peng Zhang [1,2], Bo Hu [3], Yang Gao [1,2] and Yidan Si [1,2]

1 Key Laboratory of Radiometric Calibration and Validation for Environmental Satellites, National Satellite Meteorological Center (National Center for Space Weather), China Meteorological Administration, Beijing 100081, China
2 Innovation Center for FengYun Meteorological Satellite (FYSIC), Beijing 100081, China
3 Ningbo Meteorological Burea, Ningbo 315012, China
* Correspondence: chenlin@cma.gov.cn

**Abstract:** The new-generation FengYun geostationary meteorological satellite has a high spatial and temporal resolution, which is advantageous in environmental assessments and air pollution monitoring. This study researched the ground-level particulate matter concentration estimation, based on satellite-observed radiations. The radiation of ground-level particulate matter is separate from the apparent radiation observed by satellites. The positive correlation between PM2.5 and PM10 is also considered to improve the accuracy of inversion results and the interpretability of the estimation model. Then, PM2.5 and PM10 concentrations were estimated synchronously every 5 min in mainland China based on FY-4A satellite directly observed radiations. The validation results showed that the improved model estimated results were close to the ground site measured results, with a high determination coefficient ($R^2$) (0.89 for PM2.5, and 0.90 for PM10), and a small Root Mean Squared Error (RMSE) (4.69 $\mu g/m^3$ for PM2.5 concentrations, and 13.77 $\mu g/m^3$ for PM10 concentrations). The estimation model presented a good performance in PM2.5 and PM10 concentrations during typical haze and dust storm cases, indicating that it is applicable in different weather conditions and regions.

**Keywords:** FengYun-4 geostationary meteorological satellite; machine learning; PM2.5; PM10

## 1. Introduction

The ground-level particulate matter with a diameter of less than 2.5 µm (PM2.5) and 10 µm (PM10) can stay in the atmosphere for a long period and transmit over a long distance. The atmospheric pollution caused by ground-level particulate matter has become a prominent environmental and public health problem [1–3]. The PM2.5 and PM10 concentration at ground level, as the basic indicator for air quality, provides references for air quality evaluation. Moreover, the ground-level particulate matter could also accelerate the material deterioration of buildings and other structures as well as objects of cultural heritage [4,5]. Therefore, the monitoring of PM2.5 and PM10 concentration at ground level is critical to air pollution control [6].

The ground-based monitoring sites provide the most accurate measurement. However, the ground monitoring sites only measure the PM2.5 and PM10 concentrations in a point-like distribution; therefore, it is difficult to provide a particulate matter observation at a high temporal and spatial resolution over full-country coverage. By contrast, satellite-based remote sensing has the feature of high temporal and spatial resolution observation in a wide range of coverage. Then, the dynamic changes in aerosol temporal and spatial distribution can be quickly obtained based on satellite remote sensing, which has obvious advantages in environmental and construction materials deterioration assessment [5,7,8].

The new generation of FengYun (FY-4) geostationary series meteorological satellites, with features of high spatial–temporal resolutions and more observation channels, has the unique advantage to provide a high-frequency observation of PM2.5 and PM10 concentrations at ground level over the full coverage of China.

The aerosol optical depth (AOD) represents the integrated extinction of the aerosol, which has a high correlation with particulate matter in the atmosphere. Satellite remote sensing provides AOD retrieval products and it is widely used in air quality evaluations [9,10]. Current research mainly estimates PM2.5 and PM10 concentrations based on AOD products. Machine learning technology has the advantage of describing the complex relationship between data, and showing the strong ability to extract the essential characteristics of data-sets [11]. There are already successful cases using machine learning technology to build the relationships between PM2.5 and PM10 concentration at ground level and AOD [12–15]. Chen et al. [13] used the AOD product from the Moderate Resolution Imaging Spectroradiometer (MODIS) to estimate the PM2.5 concentration at ground level over China, and considered that the extreme gradient boosting (XGBOOST) model would improve the estimation concentration of PM2.5. Wei et al. [14] improved the estimation of PM2.5 concentration using the space–time extremely randomized tree model, based on AOD products derived from MODIS and Multi-Angle Implementation of Atmospheric Correction (MAIAC). A non-linear statistics model for the PM10 estimation was developed by Wei et al. [15], based on monitoring stations measuring PM10 concentration in Xi′an and MODIS AOD product. Therefore, the AOD-based method was successfully applied in the estimation of particulate matter concentrations in the ground-level atmosphere.

However, satellite-based AOD products have uncertainties [16–18] that are adopted in the PM2.5 and PM10 concentration estimations [19] and, therefore, cause even larger uncertainties [20]. Theoretically, the satellite observation radiation could be directly related to the particulate matter concentrations and has less uncertainty than the estimation method based on the AOD products [21]. A geo-intelligent method was developed by Shen et al. [22] to directly estimate the PM2.5 concentration from reflectance at the top of atmosphere (TOA) observed by MODIS, and the result proved the method was effective. Yan et al. [23] tried to estimate the PM2.5 concentration at ground level, based on satellite observation (MODIS) radiation directly, and the results showed that the retrieval accuracy of near-ground particle concentration was better than that based on AOD products. The above studies estimated PM2.5 and PM10 concentrations based on polar orbit satellite data. However, polar orbit satellites have limits of swath width and low revisit frequency. By contrast, the geostationary satellites have higher temporal resolutions than polar orbit satellites and observe the entire development and transmittance of the air pollutant. Wei et al. [24] used the Himawari-8/AHI (Advanced Himawari Imager) AOD product to provide hourly PM2.5 estimation data. However, the Himawari-8/AHI range cannot provide full coverage of China [20].

FY-4A is the first-launched satellite of the new generation of FY geostationary series meteorological satellites; the Advanced Geosynchronous Radiation Imager (AGRI) on FY-4A provide observations over the full coverage of China, with higher temporal resolution (5 min), which is more suitable for ground-level particulate matter monitoring over China. Moreover, as the particulate matter in the atmosphere, the concentrations of PM2.5 and PM10 showed a strong positive correlation [25,26]; considering the strong positive correlation and the estimated PM2.5 and PM10 concentration synchronously should improve the estimation results. Currently, there are few studies on the synchronous PM2.5 and PM10 estimation, based on FengYun geostationary meteorological satellite observed radiations. In this research, we improved the XGBoost model to estimate the concentrations of PM2.5 and PM10 synchronously every 5 min over mainland China, based on FY-4A/AGRI observed radiations directly. The performance of the improved model was tested by cross-validation between satellite-based estimation and ground site-measured concentrations of PM2.5 and PM10. Moreover, the estimation results in different regions and weather conditions were

evaluated by analyzing the estimated concentrations of PM2.5 and PM10 during haze and dust storm events. This paper is organized as follows.

The datasets used in this article are described in Section 2. Section 3 introduces the development of the particulate matter concentration estimation model, the data preprocessing method and the result verification in detail. The FY-4A satellite-based estimation of PM2.5 and PM10 concentration results are presented and discussed in Sections 4 and 5, respectively. Finally, the conclusions are addressed in Section 6.

## 2. Data

This research aimed to estimate the concentrations of PM2.5 and PM10 synchronously every 5 min over mainland China, based on FY-4A/AGRI observed radiations. Therefore, satellite-observed radiation, ground-level particulate matter concentration, meteorological data and land surface parameter data (a summary of datasets is listed in Table 1) were employed to develop the PM2.5 and PM10 concentration estimation model.

**Table 1.** The datasets used in this study.

| Datasets | Variables | Units | Temporal Resolution | Spatial Resolution | Data Source |
|---|---|---|---|---|---|
| The ground-level particulate matter concentration | PM2.5 | $\mu g/m^3$ | Hourly | - | CNEMC |
| | PM10 | | | | |
| Satellite observed Radiation | Reflectance (Channel 1–6) | - | 5 min (China area) | 4 km | FY-4A/AGRI |
| | Brightness Temperature (Channel 7–14) | K | | | |
| Meteorological data | Boundary Layer Height | m | Hourly | 0.25° | ERA-5 |
| | Wind | m/s | | | |
| | Integrated Water Vapor | $kg/m^2$ | | | |
| | Surface pressure | hpa | | | |
| | Temperature (16 layers, 500 hpa to 1000 hpa) | K | | | |
| | Relative humidity (16 layers, 500 hpa to 1000 hpa) | % | | | |
| Land surface parameter data | Land Surface Albedo | - | 16 Days | 0.05° | MCD43C3 |
| | Land Surface Elevation | m | - | 90 m | SRTMGL1 |
| | NDVI | - | Monthly | 0.05° | MYD13C2 |

### 2.1. FY-4A Data

FengYun-4 is the new generation of the FengYun geostationary series meteorological satellite of China, and the FY-4A satellite is the first-launched satellite of the FengYun-4 series. It launched on 11 December 2016, and the center longitude was located at 104.7°E since 25 May 2017. Both the use of navigation registration technology and three-axis stabilization enhancement technology made the data of FY-4A have a higher navigation registration accuracy and observation efficiency [27]. The FY-4A satellite has four instruments [28]; among these instruments, AGRI is the primary load of the FY-4A satellite with more spectral channels (Table 2 shows the spectral and spatial resolution of FY-4A/AGRI) and a higher scanning efficiency (every 15 min for full disk, every 5 min for China area and every 1 min for target area), which will greatly improve PM2.5 and PM10 estimations [29]. Therefore, FY-4A has a unique advantage to provide full coverage and high-frequency PM2.5 and PM10 concentrations over China.

**Table 2.** The channel parameters (spectral and spatial resolution) of FY-4A/AGRI.

|  | Wavelength (μm) | Spatial Resolution (km) |
|---|---|---|
| Visible channels | 0.47 | 1 |
|  | 0.65 | 0.5 |
| Near-infrared visible channel | 0.83 | 1 |
| Shortwave infrared visible channels | 1.37 | 2 |
|  | 1.61 | 2 |
|  | 2.22 | 2 |
| Medium-wave infrared visible channels | 3.72 high | 2 |
|  | 3.72 low | 4 |
| Water vapor visible channels | 6.25 | 4 |
|  | 7.1 | 4 |
| Thermal infrared visible channels | 8.5 | 4 |
|  | 10.8 | 4 |
|  | 12.0 | 4 |
|  | 13.5 | 4 |

In this study, we used the reflectance and brightness temperature (derived from all visible and infrared channels in Table 2) at TOA observed by FY-4A/AGRI, instead of the AOD product, to directly estimate the PM2.5 and PM10 concentration. The implementation of de-cloud processing was based on the CLoud Mask (CLM) product to avoid cloud impacts on the PM estimation. To estimate the concentrations of PM2.5 and PM10 in the dust storm region, we used the Dust Storm Detection (DSD) product to detect the dust storm region. Then, the concentration of PM2.5 and PM10 over clear-sky, dust storm and haze regions could be estimated separately based on FY-4A/AGRI observations.

The FY-4A/AGRI observed reflectance and the brightness temperature of L1 data and CLM and DSD products were open-accessed from the National Satellite Meteorological Center (NSMC) (http://satellite.nsmc.org.cn (accessed on 3 January 2023)). The TOA radiation data and CLM and DSD products from the FY-4A satellite had a 4 km spatial resolution and 5 min time resolution, with a temporal range covering 2021 and 2022.

### 2.2. The Ground Level PM2.5 and PM10 Concentration Monitoring Data

The ground-based site-measured concentrations of PM2.5 and PM10 data were accessed from the China Environmental Monitoring Center (CNEMC)'s website (http://www.cnemc.cn (accessed on 10 January 2023)). The ground-based sites over mainland China were used to measure and collect air quality data, which included the hourly monitoring of the data of the PM2.5 and PM10 concentrations. The ground-based site-measured air quality data were quality-controlled following China's National Ambient Air Quality Standard (CNAAQS) to improve the data accuracy [13]; the uncertainty of the particulate matter concentration measurement is less than 5 μg/m$^3$ [30].

In this study, we collected the site-measured concentrations of PM2.5 and PM10 from 2027 ground-based sites over mainland China; the temporal ranges of the data were from 2021 to 2022. Figure 1 shows the distribution map of the ground-based sites and the research area across mainland China. We conducted quality control on hourly PM2.5 data to remove missing values and severe data outliers.

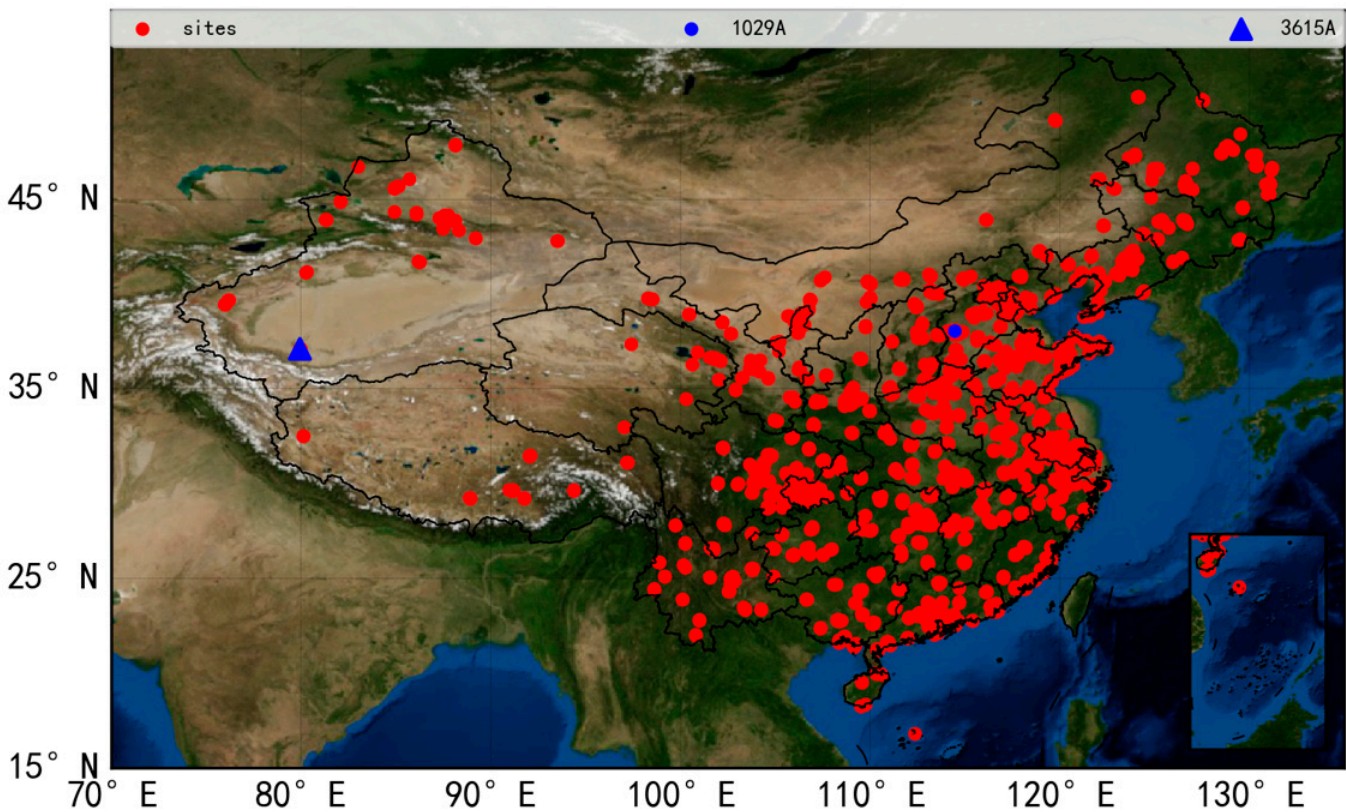

**Figure 1.** The research area (mainland China) and the distribution map of ground-based sites.

### 2.3. Meteorological Data

The distribution of the ground-level PM concentration not only has seasonal variations, but also temporal and spatial fluctuations caused by meteorological factors [31–33]. Therefore, the meteorological factors significantly impact the estimation and predictions of PM2.5 and PM10. Previous studies demonstrated that boundary layer height, wind, water vapor, surface pressure, temperature and relative humidity greatly impact the formation and dissipation of PM2.5 and PM10 [24,34,35].

The meteorological factors, including boundary layer height, wind, water vapor, surface pressure, the profile of temperature and relative humidity data, were from the fifth generation of atmospheric reanalysis of the global climate (ERA-5) data from European Centre for Medium-range Weather Forecasts (ECMWF) (https://cds.climate.copernicus.eu/ (accessed on 10 January 2023)) and were collected in this research. The ERA-5 data were hourly grided in a spatial resolution of $0.25° \times 0.25°$.

### 2.4. Land Surface Parameters Data

The apparent radiation observed by a satellite was from the land's surface and atmosphere, while the ground-level particles matters were mainly below the boundary layer [33,36]. Therefore, separating the radiation of ground-level particulate matters from the apparent radiation observed by satellites was important when estimating the concentrations of PM2.5 and PM10. The land surface parameters were helpful for the land surface and atmosphere radiation separation.

In this study, we used the land surface albedo, land surface elevation and Normalized Difference Vegetation Index (NDVI) data as the land surface parameters to input into the estimation model to estimate the concentrations of PM 2.5 and PM10 at ground level.

The surface radiation characteristics were very important to separate the radiation of land from the apparent radiation observed by satellites. In this research, we used the MODIS Collection6 albedo product dataset (MCD43C3) [37–39] to provide high-quality land surface reflectance and albedo data over various types of land surfaces using the

anisotropy retrievals algorithm [40–43]. The shortwave white-sky albedo (WSA) and black-sky albedo (BSA) from the MCD43C3 product were used to obtain the SW broadband (0.3–5.0 μm) land surface albedo [44,45].

The surface elevation was obtained from the Shuttle Radar Topography Mission (SRTM) Digital Elevation Model (DEM) product. The SRTM provided high-resolution DEM that was high quality and openly accessible [45]. In this study, we used DEM from Version 3.0 SRTM Global 1 arc second product (SRTMGL1).

NDVI can be used to represent the land cover and vegetation condition. The MYD13C2 product provided monthly cloud-free NDVI in 0.05-degree grid, based on AQUA/MODIS observations. In this study, NDVI from MYD13C2 Version 6.1 product was used as a proxy for the land cover data to estimate the concentration of PM2.5 and PM10 at ground level.

The SRTMGL1, MCD43C3 and MYD13C2 product data can be obtained from Aeronautics and Space Administration (NASA) (https://search.earthdata.nasa.gov/ (accessed on 10 January 2023)).

## 3. Methods

### 3.1. The PM2.5 and PM10 Concentration Estimation Model

XGBoost is an ensemble machine learning algorithm using the Gradient Boosting framework based on the boosting algorithm. XGBoost solved many data science problems efficiently and accurately. Therefore, XGBoost is widely used in many fields [46]. There are also many successful cases of particulate matter concentration estimation based on the XGBoost model [13,47–50]. These results proved that XGBoost outperformed various statistical models. Therefore, XGBoost was suitable for the application of the particulate matter concentration estimation.

In this research, the positive correlation between PM2.5 and PM10 concentration was considered, and the XGBoost model was improved and rebuilt to estimate the concentrations of PM2.5 and PM10 synchronously. Figure 2 shows the procedures of the estimation model, which includes the following three steps:

Step 1 was data integration. Firstly, the satellite data, ground site-measured PM concentration data, meteorological data and land surface parameters data were subject to data quality control and temporal–spatial matching. All kinds of data (summary of datasets listed in Table 1) were sampled from ground site locations (the nearest grid data from ground sites) to obtain hourly (temporal resolution of ground site measurements) matched datasets. Then, the matched dataset was normalized, and the matched dataset was also divided into the training set, testing set and validation set for model training, testing and result validating, respectively. In this study, the matched dataset in 2021 was randomly and uniformly assigned, training sets were composed by 80% of the matched dataset in 2021, and testing sets were composed by 20% of the matched dataset in 2021.

Step 2 was model development. Parameter tuning was essential to improve the model and achieve an optimal performance. Therefore, we evaluated the model based on the testing set and the parameter optimized in training. Then, the model parameters max_depth = 12 (maximum depth of the tree), eta = 0.06 (learning rate), n_estimators = 160 (number of gradient boosting trees), subsample = 0.8 (sampling rate of training samples), objective = reg:squarederror (specify the learning task and the corresponding learning objective) were used in the study for PM2.5 and PM10 estimation. The max_depth value indicates the complex of the model, increasing max_depth would make the model more complex and more likely to overfit. The eta value indicates the step size of the model, the shrinkage of eta would update to prevents overfitting, and eta shrinks the feature weights to make the boosting process of the model more conservative. Gradient boosting is fairly robust to over-fitting so a large n-estimators value usually results in better performance. The subsample value controls the sampling rate of training data, setting it to 0.8 means that model would randomly sample 80% of the training set data, and this would prevent overfitting. The objective defined the "specify the learning task" and the corresponding learning objective, reg:squarederror indicated regression with squared loss. Moreover, the

positive correlation between PM2.5 and PM10 was considered, and the XGBoost model was rebuilt to a multi-output model by using MultiOutputRegressor from Scikit-learn library. This strategy extended regressors of XGBoost to support multi-target regression, it consisted of fitting one regressor per target, and established the connection between targets. Therefore, the model was improved to estimate the PM2.5 and PM10 concentrations at ground level, synchronously.

Step 3 was model application. FY-4A/AGRI data, meteorological data and land surface parameters data were each projected and interpolated (using the bilinear interpolation method) to a 4 km resolution used for the improved estimation model input, and the improved estimation model was applied to estimate PM2.5 and PM10 concentrations synchronously over mainland China every 5 min.

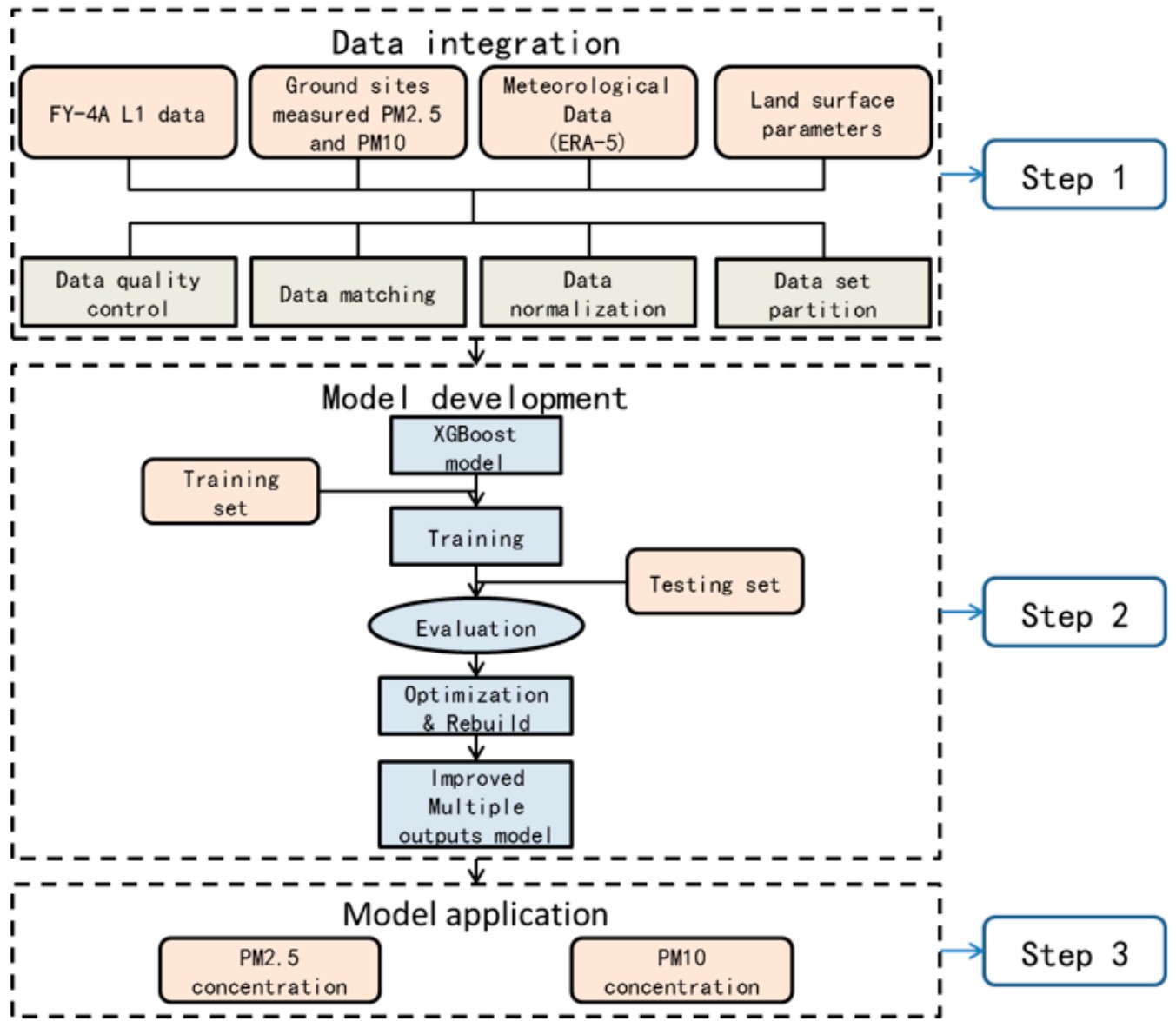

**Figure 2.** The flowchart of the estimation model development and application.

### 3.2. Results Verification Method

The performance of the improved estimation model was evaluated using statistical results, including the coefficient of determination ($R^2$), the Root Mean Square Error (RMSE), Mean Error (ME) and the Agreement index ($A_{index}$).

$$R^2 = 1 - \frac{\sum_i^N (y_i - y_{estimate})^2}{\sum (y_i - \overline{y})^2} \tag{1}$$

$$RMSE = \sqrt{\frac{\sum_i^N (y_i - y_{estimate})^2}{N}} \tag{2}$$

$$ME = \frac{\sum_i^N (y_i - y_{estimate})}{N} \tag{3}$$

$$A_{index} = 1 - \frac{\sum_i^N (y_{estimate} - y_i)^2}{\sum_i^N (|y_{estimate} - \overline{y_i}| - |y_i - \overline{y_i}|)^2} \tag{4}$$

where $y_{estimate}$ represents the estimated PM2.5 and PM10 concentrations based on FY-4A/AGRI observations, $y_i$ is the ground site-measured values, $\overline{y_i}$ is the mean value of ground site-measured results and $N$ is the number of the ground site measurements.

## 4. Results

The improved XGBoost model was applied to estimate the concentrations of PM2.5 and PM10 every 5 min over mainland China based on FY-4A/AGRI observed radiations directly. In this section, the accuracy of the estimation results was tested by ground site measurements. Moreover, the concentrations of PM2.5 and PM10 during typical haze and dust storm cases were analyzed separately to test the applicability of the improved model in various weather conditions and regions.

### 4.1. Evaluation of the Estimation Model

The validation set (22,980 points, independent matched data set in 2022, 12% of integration data in 2022 were randomly and uniformly sampled) was used to evaluate the estimation model by comparing the concentrations of PM2.5 and PM10 estimation results and ground-based site-measured results.

Figure 3 shows the cross-validation of the PM2.5 concentration (Figure 3a) and PM10 (Figure 3b) concentration between satellite-based estimations and ground site-based measurements. The results show that the FY-4A satellite-based estimation PM2.5 and the PM10 concentration were highly consistent with the ground site-measured results, with a high determination coefficient ($R^2$) (0.81 for PM2.5, and 0.73 for PM10) and a small Root Mean Squared Error (RMSE) (11.58 μg/m$^3$ for PM2.5 concentration, and 23.69 μg/m$^3$ for PM10 concentration).

The difference histogram (Figure 4) shows there are few differences between satellite-based estimations and ground-based site measurements PM2.5 (mean error was −0.012 μg/m$^3$, and median error was −0.0007 μg/m$^3$) and PM10 (mean error was −0.18 μg/m$^3$, and median error was −0.0011 μg/m$^3$) concentration. The difference corresponded to the normal distribution; the errors were mainly around 0 μg/m$^3$.

The monthly mean concentrations of PM2.5 and PM10 from satellite-based estimation results were also evaluated by ground-based site-measured results (monthly mean of 2027 site measurements). Figure 5 shows the cross-validation of the monthly mean PM2.5 concentration (Figure 5a) and PM10 (Figure 5b) concentration between satellite-based estimation and ground site-based measurements in March 2022. The result shows that the FY-4A satellite-based estimated monthly mean PM2.5 and PM10 concentrations were highly consistent with the ground site-measured results, with a high determination coefficient (R$^2$) (0.89 for PM2.5, and 0.90 for PM10) and a small Root Mean Squared Error (RMSE) (4.69 μg/m$^3$ for PM2.5 concentrations, and 13.77 μg/m$^3$ for PM10 con-

centrations). The difference histogram (Figure 6) between satellite-based estimation and ground-based site-measured results shows that the difference corresponds to the normal distribution; there was also little difference between satellite-based estimation and ground-based site-measured monthly mean PM2.5 (mean error was $-0.23$ µg/m$^3$, and median error was $-0.38$ µg/m$^3$) and PM10 (mean error was $-0.006$ µg/m$^3$, and median error was $-0.93$ µg/m$^3$) concentration.

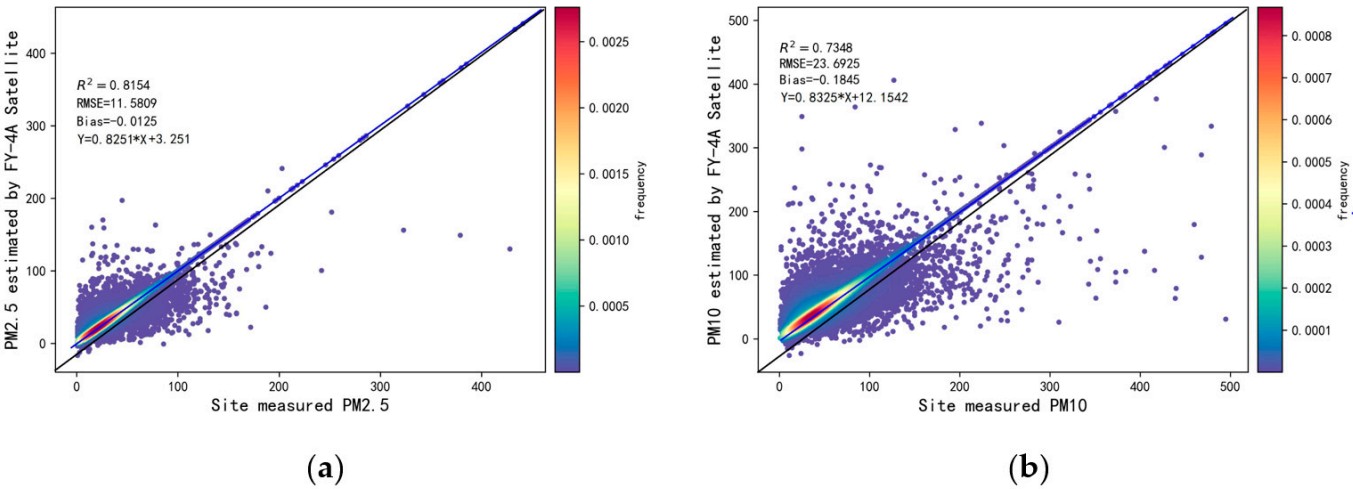

**Figure 3.** The scatter plots for cross-validation of (**a**) PM2.5 and (**b**) PM10 concentration results. The color represents the data frequency.

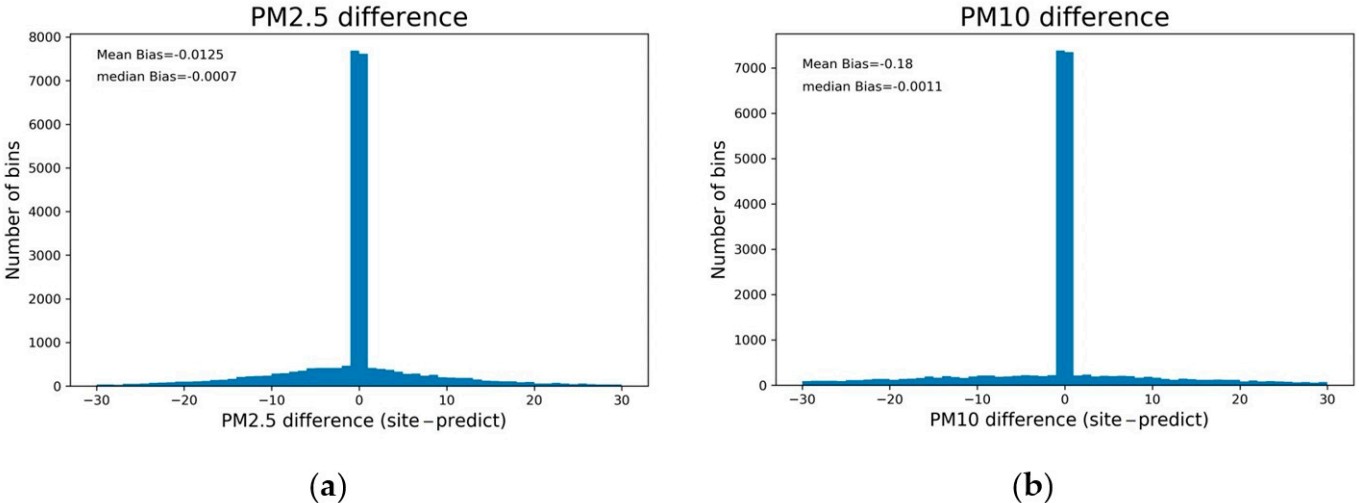

**Figure 4.** The difference histogram of (**a**) PM2.5 and (**b**) PM10 concentration between satellite-based estimation and ground-based site measurements.

The monthly mean PM2.5 and PM10 concentrations estimated by FY-4A/AGRI sampled point-like data following ground-based site locations. The distribution maps of satellite-based estimated monthly mean PM2.5 (Figure 7c) and PM10 (Figure 7d) concentrations in March 2022 were highly consistent with ground site measured results (Figure 7a,b), both results show high-value regions in North China (haze) and Xinjiang (dust). This indicates that the particulate matter concentration estimation results based on FY-4A/AGRI have a high accuracy and rationality. Some regions show the difference between ground site measurements, especially in high PM2.5 and PM10 concentration value regions over high land surface albedo regions.

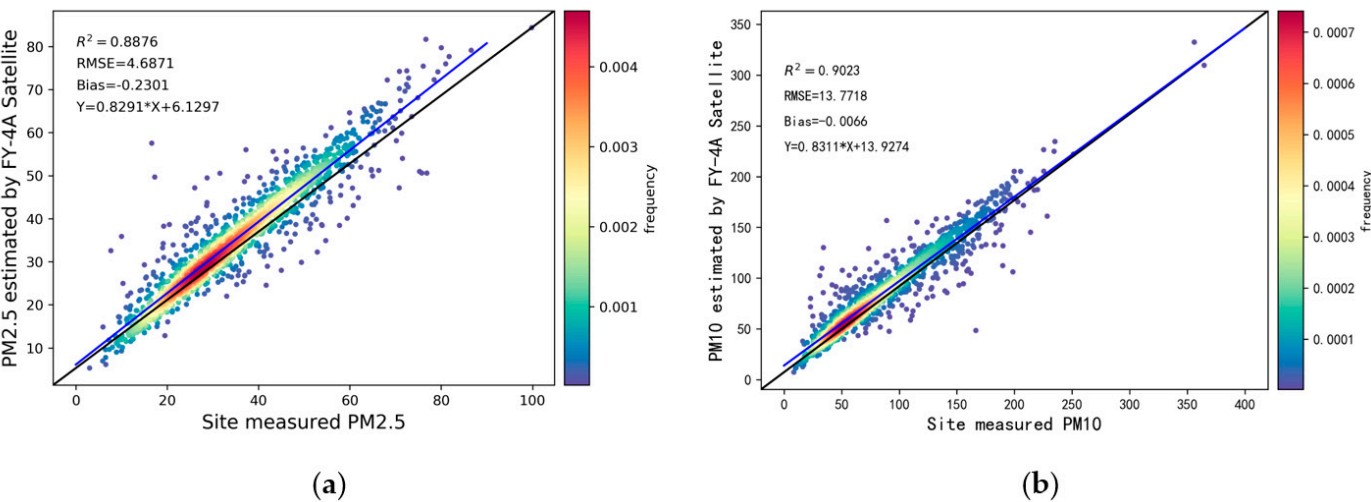

(**a**)　　　　　　　　　　　　　　　　(**b**)

**Figure 5.** The scatter plots for cross-validation of monthly mean (**a**) PM2.5 and (**b**) PM10 concentration results in March 2022. The color represents the data frequency.

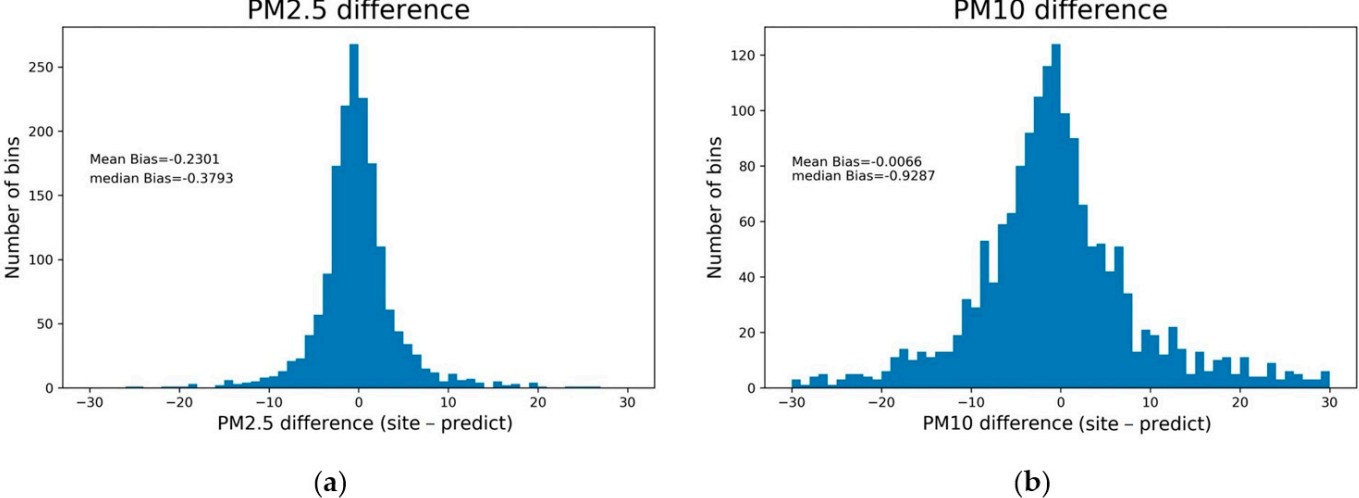

(**a**)　　　　　　　　　　　　　　　　(**b**)

**Figure 6.** The difference histogram of monthly mean (**a**) PM2.5 and (**b**) PM10 concentration between satellite-based estimation and ground-based site measurements in March 2022.

The performance of the PM2.5 and PM10 concentration estimations was also evaluated in different months. The temporal evolution results indicated the characteristics of seasonal variation. Figure 8 shows the PM2.5 and PM10 estimation results had a higher accuracy (lower RMSE) in warm seasons and a lower accuracy (higher RMSE) in cold seasons; this may have resulted from the land cover changes. The land surface albedo was quite high and was not stable due to the snow cover in cold seasons and the relationships between visible and SWIR channels over bright regions resulting in large difficulties in land surface albedo estimation [18,40–43,50]. Therefore, the ground-level particulate matter concentration estimations may have larger uncertainties in cold seasons.

*4.2. Estimation of PM2.5 and PM10 Concentration during Haze and Dust Storm Weather*

The estimation model performance of applicability in different weather conditions and regions was tested by estimating particulate matter concentrations during typical haze and dust storm cases.

The typical haze cases occurred over north China on 10 March 2022. The true color image observed by FY-4A/AGRI (Figure 9a) showed obvious haze regions over north China. The novel algorithm of haze identification was employed to detect haze regions [51];

the algorithm was built upon the spectral characteristics of different pixels: the visible band reflectance of the cloud pixels was greater than clear sky pixels, while the brightness temperature in the infrared channels was usually lower than the clear sky and haze pixels, and the brightness temperature difference was also higher than that of the clear sky and haze pixels. Then, the above characteristics were taken, and the adaptive improvements to the threshold selection were carried out for haze detection based on FY-4A/AGRI, and the PM2.5 (Figure 9b) and PM10 (Figure 9c) concentrations were estimated during the haze pollution processes. The satellite-based estimation results were highly consistent with ground-site-measured PM2.5 (Figure 9d) and PM10 (Figure 9e) concentration; both results showed a high-value center over the Hebei province and a low value in clear-sky regions.

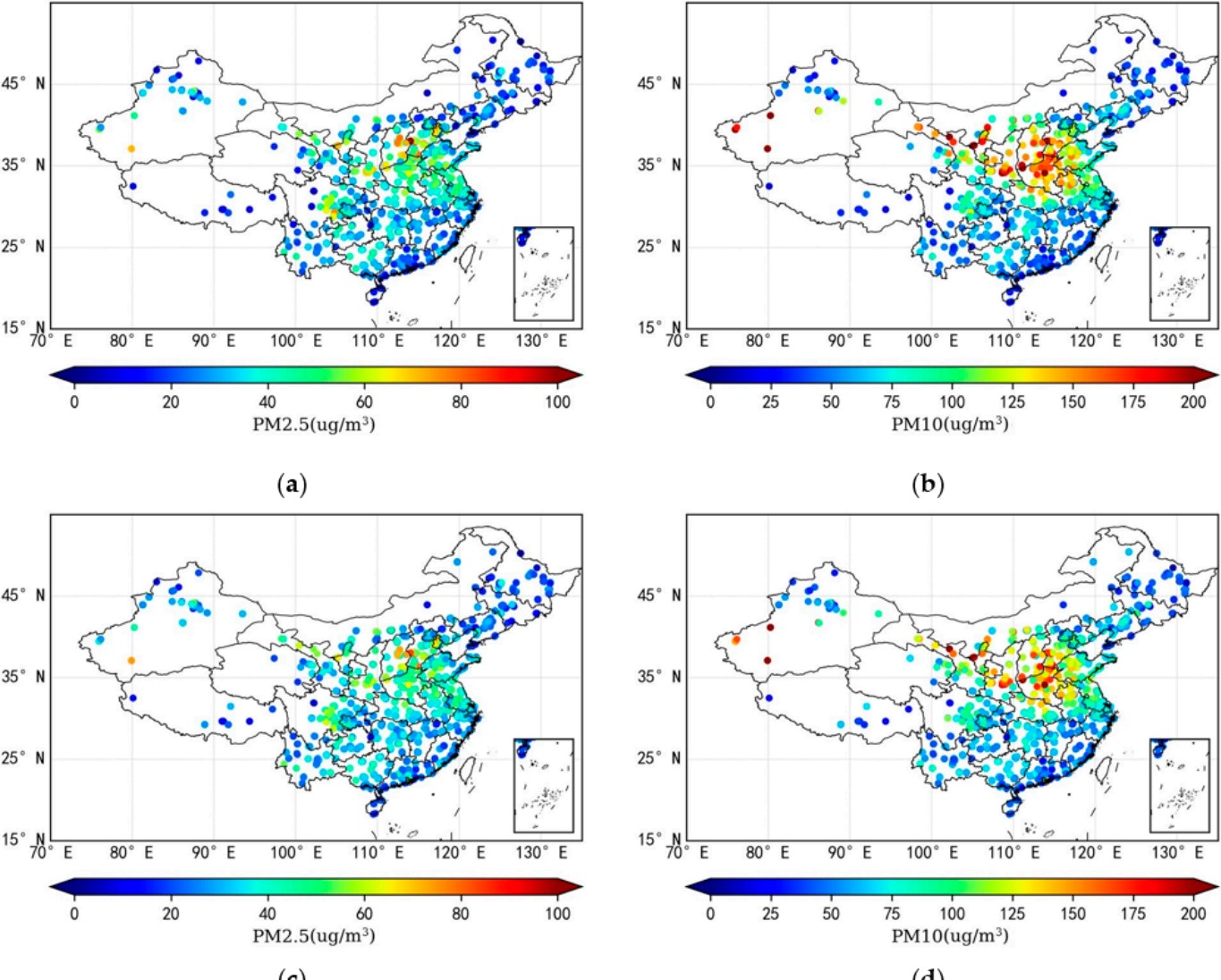

**Figure 7.** The distribution maps of ground-based site-measured monthly mean (**a**) PM2.5 and (**b**) PM10 concentrations, and satellite-based estimated monthly mean (**c**) PM2.5 and (**d**) PM10 concentrations in March 2022.

A strong dust storm occurred over the northwest of China on 20 April 2022. The dust storm was also observed by FY-4A/AGRI (Figure 10a), dust storm regions were detected by FY-4A DSD product, and PM2.5 (Figure 10b) and PM10 (Figure 10c) concentration estimated both over dust storm and clear-sky regions using the PM2.5 and PM10 concentration estimation model developed in this paper. The estimated results compared to ground-site-measured results are presented in Figure 10d,e. The satellite-based estimation results

were highly consistent with ground site-measured results; both results showed high-value regions over Xinjiang, Inner Mongolia and the northeast of China, and a low value in clear-sky regions.

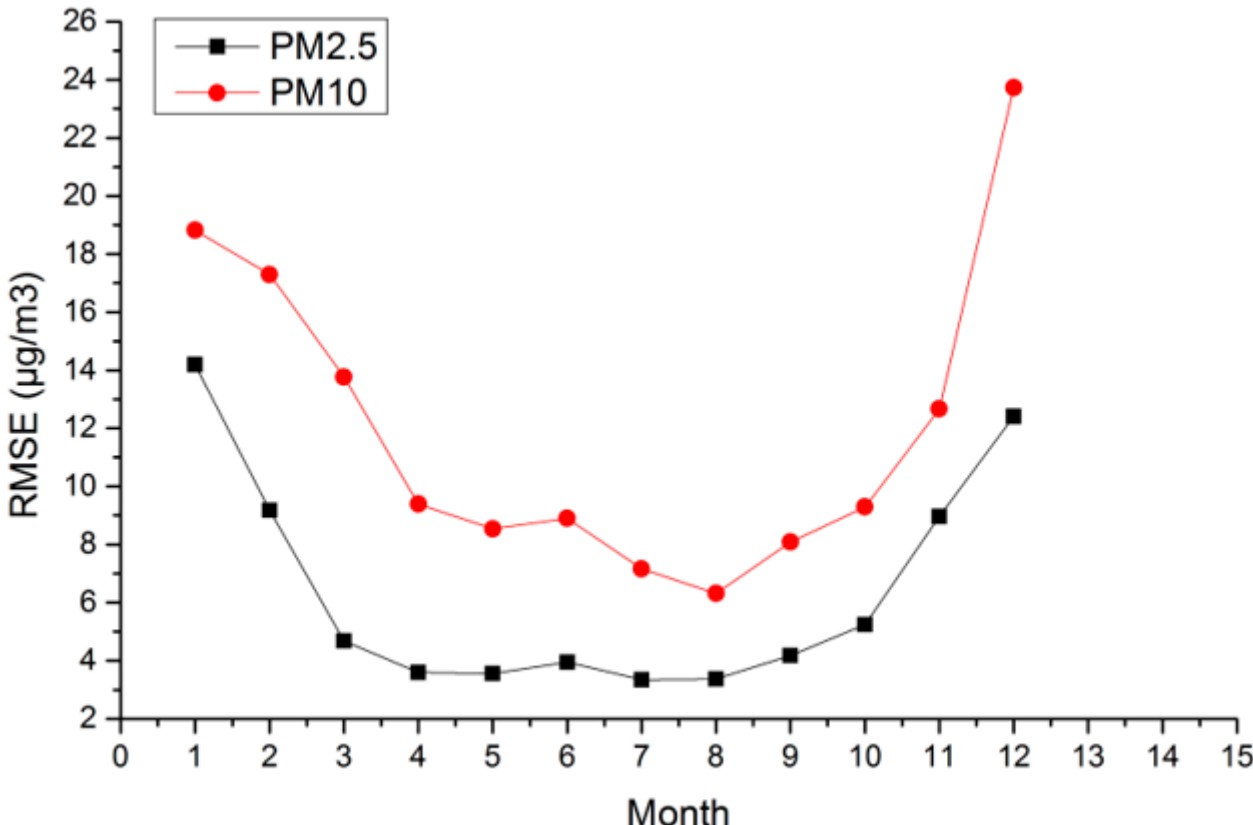

**Figure 8.** The monthly RMSE variation of FY-4A/AGRI estimated PM2.5 and PM10 concentrations.

In order to analyze the development and dissipation process of air pollution, the aerosols over Shijiazhuang were mainly considered as mainly sources of anthropogenic aerosols from biomass burning and industrial activities [52]. Hotan city is surrounded by the Taklimakan desert and the major aerosol over Hotan is dust aerosol during the dusty period [53]. Typical haze and dust storms occurred over the northwest of China on 10 March and 20 April, respectively. Therefore, we collected the air quality data measured by the ground site (PM2.5 and PM10 concentrations) in Shijiazhuang (site number 1029A, located at 114.4422°E, 38.0444°N, ● in Figure 1) and Hotan (site number 3615A, located at 79.9131°E, 37.113°N, ▲ in Figure 1) during the haze and dust storm cases that occurred on 10 March and 20 April 2022.

Figure 11 shows the time series observation and estimation of particulate matter concentrations over ground-based sites 1029A (Figure 11a) and 3615A (Figure 11b) on March 10 and April 20 separately. The result shows that concentrations of PM2.5 and PM10 estimated by FY-4A/AGRI highly agreed with the ground site-measured results in time series (with the agreement index of 0.93 and 0.79 for PM2.5, 0.96 and 0.93 for PM10 on March 10 and April 20 over ground sites 1029A and 3615A, respectively), which accurately monitor the development process of ground level particulate matters during haze and dust storm cases. The analysis and statistics of the results showed that PM2.5 accounted for about 50% of PM10 over haze regions, and about 25% over dust storm regions; this indicates that the particulate matter is mostly fine particles over haze regions and mostly coarse particles over dust storm regions.

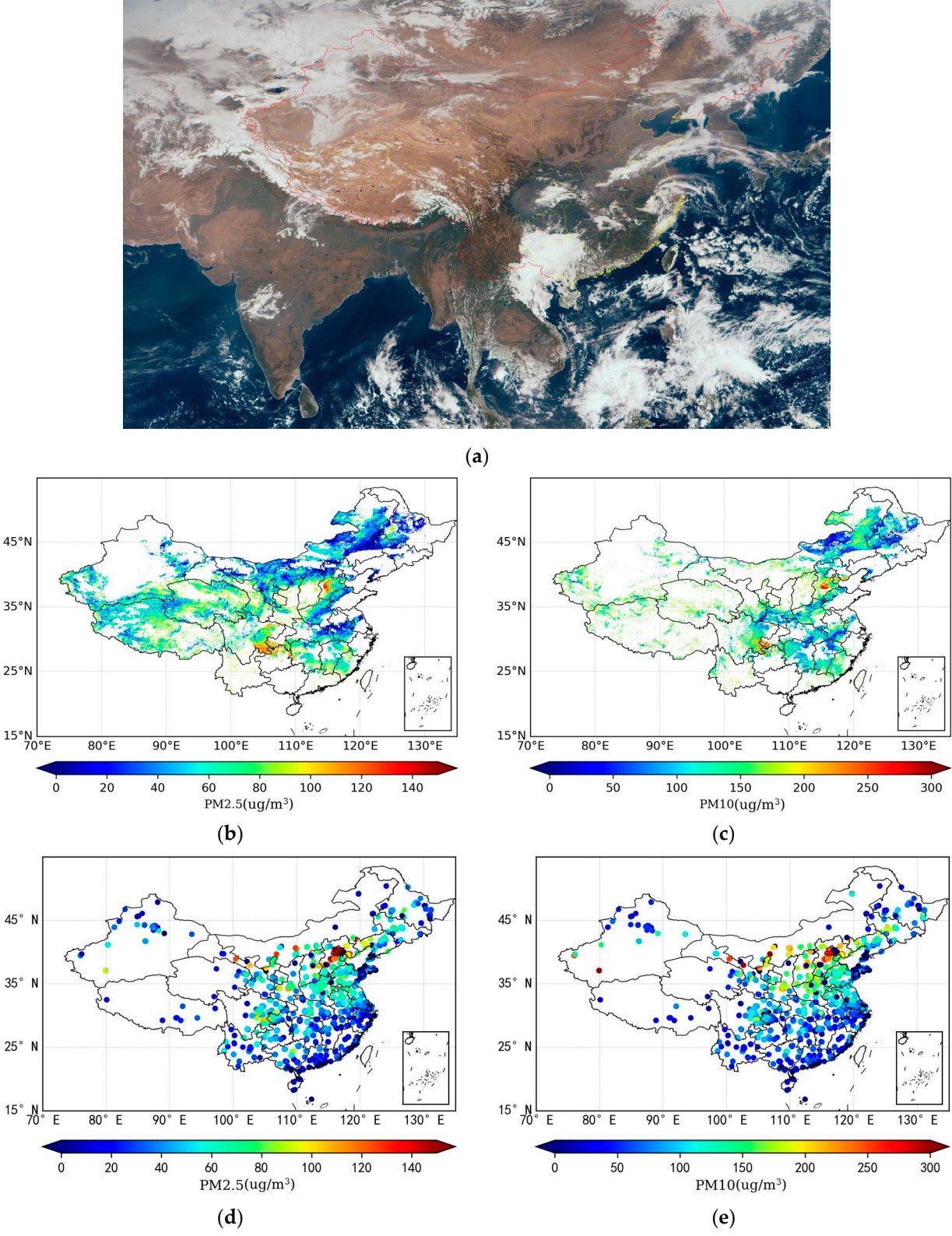

**Figure 9.** The (**a**) true color image of FY-4A/AGRI, (**b**) PM2.5 and (**c**) PM10 concentration satellite-based estimation, and ground-based site-measured (**d**) PM2.5 and (**e**) PM10 concentration during haze weather occurred on 10 March 2022.

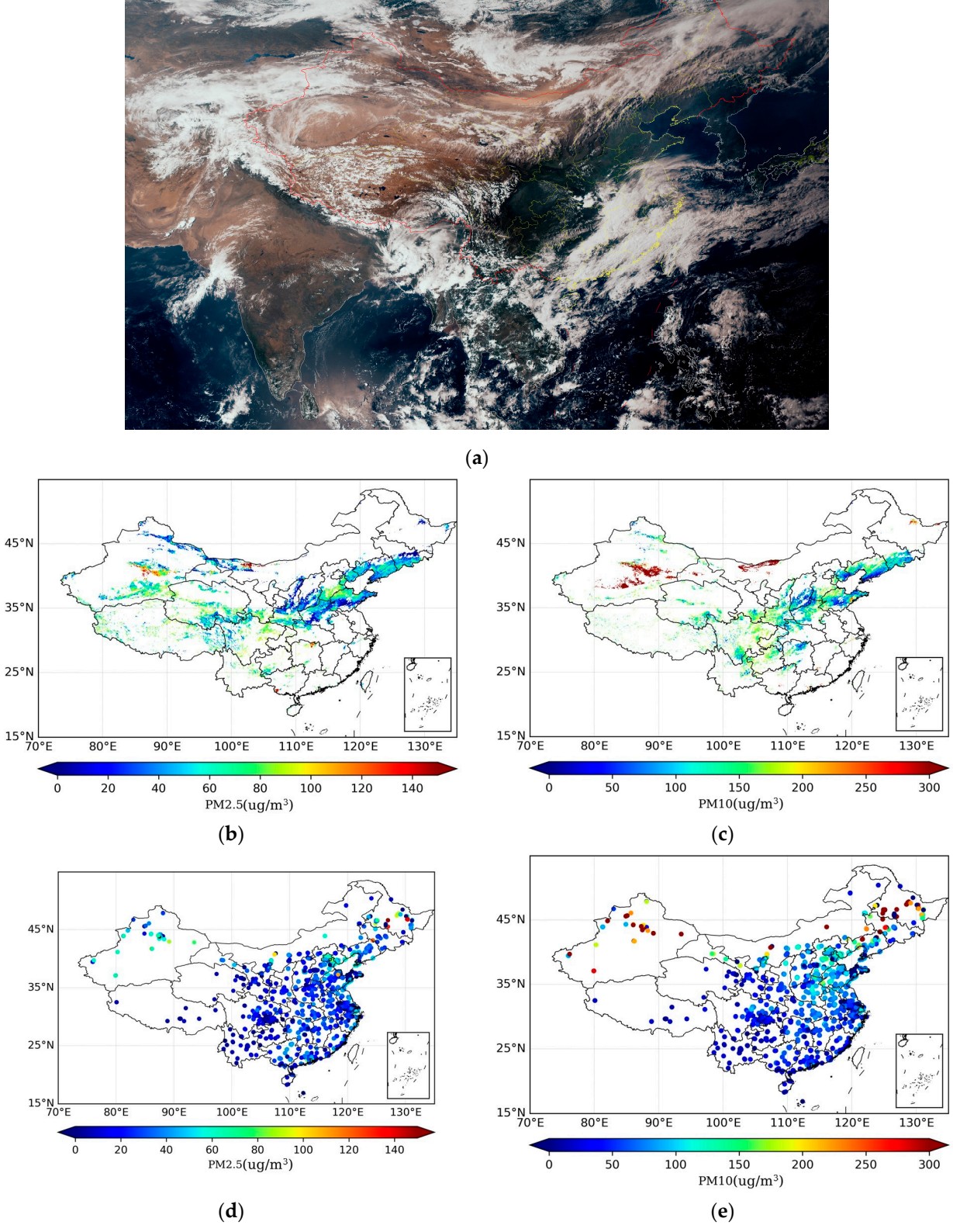

**Figure 10.** The (**a**) true color image of FY-4A/AGRI, (**b**) PM2.5 and (**c**) PM10 concentration satellite-based estimation, and ground-based site-measured (**d**) PM2.5 and (**e**) PM10 concentration during dust storm occurred on 20 April 2022.

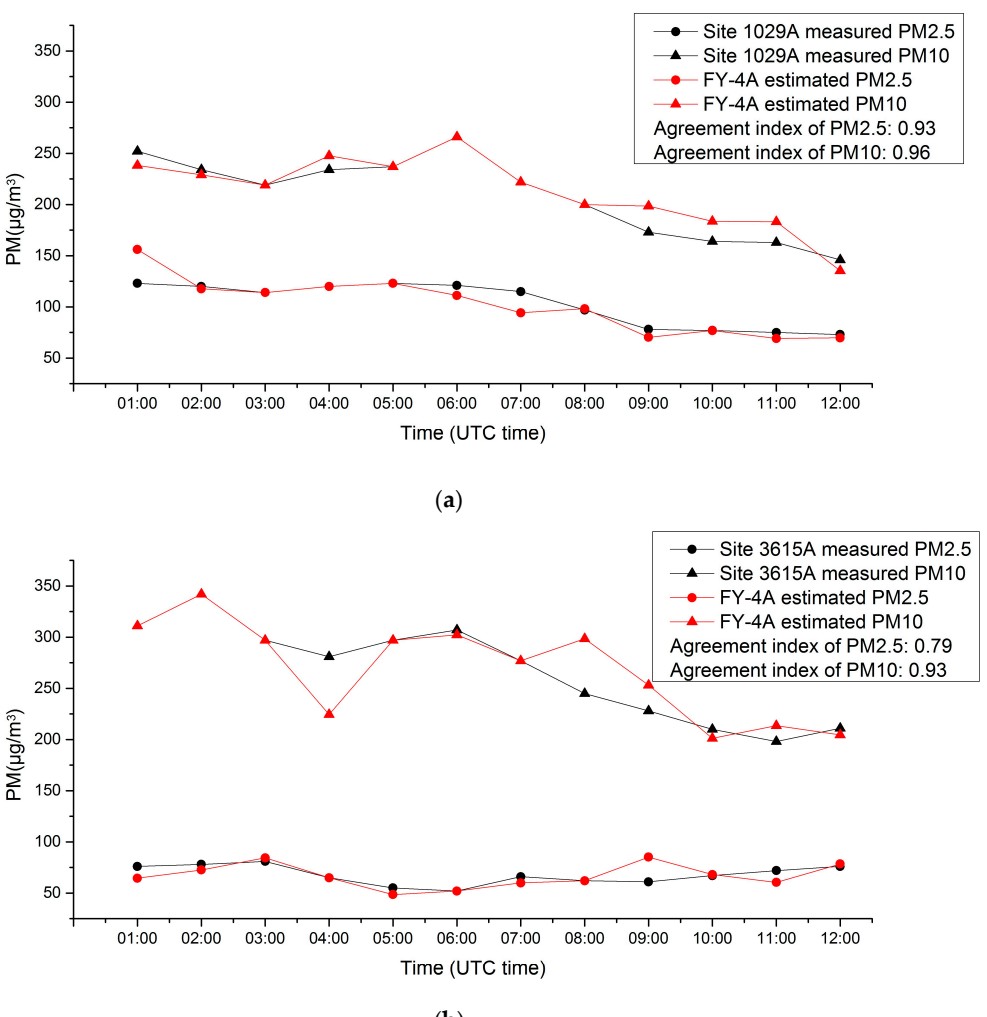

**Figure 11.** The time series observation and estimation of PM2.5 and PM10 concentrations over ground sites (**a**) 1029A and (**b**) 3615A.

The estimated results during haze and dust storm weather conformed to haze and dust aerosol characteristics; this indicates that the estimation model performed well under different weather conditions and in different regions.

## 5. Discussion

In this study, the concentrations of PM2.5 and PM10 at ground level were estimated synchronously every 5 min in mainland China based on FY-4A/AGRI directly observed radiations.

The validation results showed that the improved model of the PM2.5 and PM10 concentration estimation was close to ground site-measured results, with a high determination coefficient ($R^2$) (0.89 for PM2.5 concentration, and 0.90 for PM10 concentration) and a small Root Mean Squared Error (RMSE) (4.69 $\mu g/m^3$ for PM2.5 concentration, and 13.77 $\mu g/m^3$ for PM10 concentration). The accuracy of the PM2.5 and PM10 concentration estimation result based on FY-4A/AGRI in this paper was comparable to other studies (Table 3) [13–15,22–24,47,54].

**Table 3.** The ground-level particulate matter concentration estimated from different studies.

| Research | Study Area | Model | Temporal Resolution | Spatial Resolution | $R^2$ | RMSE (μg/m³) |
|---|---|---|---|---|---|---|
| Wei et al. (2016) [15] | Xi'an | A nonlinear model | Daily | 10 km | 0.79 (PM10) | 11.7 (PM10) |
| Li et al. (2017) | Wuhan Urban Agglomeration | Deep Belief Network (DBN) | Daily | 1 km | 0.87 (PM2.5) | 9.89 (PM2.5) |
| Chen et al. (2019) [13] | Mainland China | XGboost | Daily | 3 km | 0.86 (PM2.5) | 14.98 (PM2.5) |
| Gui et al. (2020) [47] | Mainland China | XGboost | Hourly | 0.5° × 0.625° | 0.80 (PM2.5) | 14.75 (PM2.5) |
| Yan et al. (2021) [23] | Mainland China | Spatial-Temporal Interpretable Deep Learning Model (SIDLM) | Daily | 250 m (PM2.5) | 0.62 (PM2.5) | 16.01 (PM2.5) |
| | | | | 3 km (PM2.5) | 0.66 (PM2.5) | 15.96 (PM2.5) |
| | | | | 10 km (PM2.5) | 0.70 (PM2.5) | 15.30 (PM2.5) |
| Wei et al. (2020) [14] | Mainland China | Space–Time Extremely randomized Trees (STET) | Daily | 1 km | 0.89 (PM2.5) | 10.35 (PM2.5) |
| Wei et al. (2021) [24] | Eastern China | Light Gradient Boosting Machine (LightGBM) | Hourly | 5 km | 0.98 (PM2.5) | 3.23 (PM2.5) |
| Mao et al. [54] | Mainland China | Random Forest model | Hourly | 4 km | 0.88–0.95 (PM2.5) | 5.02-12.43 (PM2.5) |
| This study | Mainland China | Improved XGboost | 5 min | 4 km | 0.89 (PM2.5) | 4.69 (PM2.5) |
| | | | | | 0.90 (PM10) | 13.77 (PM10) |

The estimation model presented a good performance during the typical haze and dust storm cases, and the results correctly presented the actual distribution and variation in ground-level particulate matters, indicating that it was applicable in different weather conditions and regions. Moreover, the concentrations of PM2.5 and PM10 were estimated in 5 min intervals, which greatly improved the temporal resolution of the PM2.5 and PM10 estimation in the current research (hourly) [24,48,54]. The high-temporal-resolution observation was important in the monitoring of PM2.5 and PM10 and the analysis of the generation and transmission of the particulate matter in the atmosphere.

There were still several potential limitations in this research. Although the particulate matter concentrations estimated by FY-4A/AGRI were highly consistent with ground-site-measured results in the spatial and time series, there were still differences with ground site measurements (Figures 8 and 11). To solve this problem, the advantage of the high spatial–temporal resolution observation from FY-4A/AGRI should be fully taken, and the spatial–temporal self-correlation of ground-level particulate matter should be considered for future improvements in PM2.5 and PM10 estimation.

## 6. Conclusions

In this research, we improved the XGBoost model, estimating the concentrations of PM2.5 and PM10 synchronously over mainland China every 5 min, based on FY-4A/AGRI directly observed radiation, which greatly improved the temporal resolution of PM2.5 and PM10 estimation in the current studies (hourly). The accuracy of the particulate matter concentration estimation results was tested by ground site measurements, and the concentrations of PM2.5 and PM10 were estimated during typical haze and dust storm cases. The validation results showed that the improved model estimated monthly mean concentrations of PM2.5 and PM10 were close to the ground site-measured results, with a high determination coefficient ($R^2$) (0.89 for PM2.5, and 0.90 for PM10) and a small Root

Mean Squared Error (RMSE) (4.69 μg/m$^3$ for PM2.5 concentrations, and 13.77 μg/m$^3$ for PM10 concentrations). The results correctly presented the actual distribution and variation of the ground-level particulate matter, indicating that it was applicable under different weather conditions and in different regions.

Future research on ground-level particulate matter should consider the spatial–temporal self-correlation of particulate matter to improve the PM2.5 and PM10 estimation, achieve the goal of accurate observation of the ground-level particulate matter concentration with a high temporal and spatial resolution, and provide a reference for air pollution control and construction materials deterioration.

**Author Contributions:** Conceptualization, L.C. and P.Z.; methodology, L.C., L.T. and Y.S.; validation, L.T., B.H. and Y.G.; formal analysis, L.T. and Y.G.; investigation, L.C. and L.T.; data curation, L.T., B.H. and Y.S.; writing—original draft preparation, L.T. and L.C.; writing—review and editing, L.C. and P.Z.; visualization, L.T. and Y.S.; supervision, L.C. and P.Z.; project administration, L.C. and P.Z.; funding acquisition, L.C., L.T. and Y.S. All authors have read and agreed to the published version of the manuscript.

**Funding:** This research was funded by the National Key Research and Development Program of China (grant number 2021YFB3901000, 2021YFB3901005), the National Natural Science Foundation of China (grant number 41875133) and the Beijing Municipal Natural Science Foundation (grant number 8214065).

**Acknowledgments:** The authors would like to thank the following for their support: FY-4A L1 data and products (CLM and DSD) from NSMC, PM2.5 and PM10 concentration measured data from the CNEMC; meteorological data (boundary layer height, wind, water vapor, surface pressure, temperature and relative humidity) from ECMWF and land surface parameters data (SRTMGL1, MCD43C3 and MYD13C2 products) from NASA.

**Conflicts of Interest:** The authors declare no conflict of interest.

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
