# Peer review of "The Ground-Level Particulate Matter Concentration Estimation Based on the New Generation of FengYun Geostationary Meteorological Satellite"

_remotesensing, doi:10.3390/rs15051459_

Round 1
Reviewer 1 Report
The manuscript studies the ground-level particulate matter concentration estimation based on the FengYun-4A satellite observed radiations using XGBoost method. It is meaningful for air pollution monitoring. The models are improved by separation of the radiation of ground-level particulate matter from the apparent radiation observed by satellites. However, the paper contains unclear descriptions and interpretations. The results are often presented without proper clarity. To summarize, in my view, to make a case for publication, the authors need a major revision.
Major Comments
P6L198-201: According to your description, XGBoost has been widely used for particulate matter concentration estimation in previous studies. What’s the advantage of the improved model? Is your improved model more accurate than previous studies? Please add a comparison and discussion between your improved models and models in previous studies.
P5L202 “The positive correlation between PM2.5 and PM10 concentration is considered” : According to Figure 8, no clear correlation between PM2.5 and PM10 is found. Please explain how to get the positive correlation between PM2.5 and PM10 concentration. Is the correlation between them significant?
P6L211 There are various data with different spatial resolutions. According to Fig. 1, the ground sites are very sparse in the western regions of China. The ERA5 data has only a 0.25x0.25deg spatial resolution. How could you obtain 4 km resolution matched data with the interpolation?
P5L223-225 How do you get the improved model to estimate PM2.5 and PM10 concentrations every 5 minutes based on FY-4A/AGRI observed radiations directly? According to Fig. 2 the input data for the XGBoost include hourly FR-4AL1 data, ground sites, meteorological data and land surface parameters. They are inconsistent. Please add details for the input and output data for your model.
P6L213-214 Please explain the strategy to divide the training set, testing set and validation set.
P7L241-242 “The accuracy of the estimation results were tested by ground sites measurements”. If the ground site measurement is used as input of your model, it can not be used for validation!
P7L246-247 The data sample for validation is 22980 points. According to your description, you use 2027 ground-based sites over the mainland of China with hourly data. And one year samples should be 17756520 and the matched data in 2022 is only 0.129%. There are only 10 samples per station on average. Please explain the reason for it. Please add a statistic for each station and the number of data samples in 2021 used for model training.
Specific Comments
P3L123-125 “CLM(CLound Mask product)product” product is redundant.
P5l178 What is “SW”?
Figure 4 The bin interval of the histogram in the figures is unclear.
Figure 5-6 What’s the white color mean in the figures? Why are the regions with white color in (b) and (d) different? The value of 20 and 40 for PM2.5 and the value 50 and 100 for PM10 are difficult to distinguish in the figures.
P11L293-399 please explain why you choose the Shijiazhuang and Hotan station for the analysis.
P13L342 ‘Altough’ to ‘Although’
Author Response
Responses to the comments from Reviewer1
Dear Reviewer,
Thank you for reviewing our manuscript. The valuable comments and suggestions have helped us to improve the paper. Please find below the comments in blue italics and our responses in black, the changes were highlighted in the revised manuscript.
Comments and Suggestions for Authors
The manuscript studies the ground-level particulate matter concentration estimation based on the FengYun-4A satellite observed radiations using XGBoost method. It is meaningful for air pollution monitoring. The models are improved by separation of the radiation of ground-level particulate matter from the apparent radiation observed by satellites. However, the paper contains unclear descriptions and interpretations. The results are often presented without proper clarity. To summarize, in my view, to make a case for publication, the authors need a major revision.
Reply: We thank the reviewer for the valuable comments that have helped us to improve our manuscript. Following these comments and suggestions, we have taken a lot of efforts to make the writing more concise and clear.
Major Comments
- P6L198-201: According to your description, XGBoost has been widely used for particulate matter concentration estimation in previous studies. What’s the advantage of the improved model? Is your improved model more accurate than previous studies? Please add a comparison and discussion between your improved models and models in previous studies.
Reply: Thank you for the valuable suggestion, we have added a comparison and discussion between improved models and models in previous studies in the revised manuscript. Please see Table 3 and Line 391-393 in Page 15-16 of the revised manuscript.
XGBoost is an optimized distributed gradient boosting library, which based on the boosting algorithm. It implements machine learning algorithms under the Gradient Boosting framework, with the advantages of high efficiency, flexibility and portability. XGBoost has solved many data science problems in a fast and accurate way, and is widely used in data mining, recommendation systems and other fields. There are also many successful cases of PM2.5 and PM10 concentration estimation based on XGBoost model [1,2], these results proved XGBoost is outperform in various statistical models. Therefore, the XGBoost algorithm was selected in this study for PM2.5 and PM10 estimation. The parameter of the model was optimized in training, the positive correlation between PM2.5 and PM10 is considered, and XGBoost model was rebuilt to an Multi-output model for estimating PM2.5 and PM10 concentrations in ground level synchronously.
The improved model presents good performance in PM2.5 and PM10 concentration estimating. The ground-level particulate matter concentration estimated based on XGBoost model from different studies are listed in Table 1, the comparison of the results shows the ground-level particulate matter concentration estimated in this paper has higher accuracy.
Table 1. The ground-level particulate matter concentration estimated from different studies.
|
Research |
Study area |
R2 |
RMSE (μg/m3) |
|
Chen et al (2019) [1] |
mainland of China |
0.86 for PM2.5 |
14.98 for PM2.5 |
|
Gui et al (2020) [2] |
mainland of China |
0.80 for PM2.5 |
14.75 for PM2.5 |
|
This study |
mainland of China |
0.89 for PM2.5 |
4.69 for PM2.5 |
|
0.90 for PM10 |
13.77 for PM10 |
- P5L202 “The positive correlation between PM2.5 and PM10 concentration is considered” : According to Figure 8, no clear correlation between PM2.5 and PM10 is found. Please explain how to get the positive correlation between PM2.5 and PM10 concentration. Is the correlation between them significant?
Reply: PM2.5 and PM10 are the particulate matters in different diameters, previous studies have proved the strong positive correlation between concentrations of PM2.5 and PM10 [3,4]. Therefore, the positive correlation between PM2.5 and PM10 is considered in estimation, and the model was improved to Multi-output model for estimating PM2.5 and PM10 concentrations synchronously. Figure 8 shows the particulate matter concentrations during typical haze case, both PM2.5 and PM10 results show a high-value center over haze regions and a low values in clear-sky regions. The results shows PM2.5 accounting for about 50% of PM10 over haze region, and about 25% over dust storm region, this indicate that the particulate matters are mostly fine particles over haze regions, and mostly coarse particles over dust storm regions.
- P6L211 There are various data with different spatial resolutions. According to Fig. 1, the ground sites are very sparse in the western regions of China. The ERA5 data has only a 0.25x0.25deg spatial resolution. How could you obtain 4 km resolution matched data with the interpolation?
Reply: We took different data matching methods in data integration and model application.
In data integration, all kinds of data (FY-4A/AGRI data, Meteorological data, and Land surface parameters data) were sampled following ground sites location (nearest grid data from ground sites) to get hourly (temporal resolution of ground sites measurements) matched dataset. Then the matched data set was normalized, and the matched data set also divided into training set, testing set and validation set for model traning, testing and result validating respectively.
In model application, both FY-4A/AGRI data, Meteorological data, and Land surface parameters data were projected and interpolated (using bilinear interpolation method) to 4 km resolution used for improved estimation model input, and the improved estimation model was applied to estimate PM2.5 and PM10 concentrations synchronously over mainland of China in every 5 minutes.
We have added the processing details in the revised manuscript, please see Line 220-227 and Line 237-241 in Page 7 of the revised manuscript.
- P5L223-225 How do you get the improved model to estimate PM2.5 and PM10 concentrations every 5 minutes based on FY-4A/AGRI observed radiations directly? According to Fig. 2 the input data for the XGBoost include hourly FR-4AL1 data, ground sites, meteorological data and land surface parameters. They are inconsistent. Please add details for the input and output data for your model.
Reply: FY-4A/AGRI has higher scanning efficiency (every 15 minute for full disk, every 5 minute for China area, and every 1 minute for target area), it has unique advantage to provide full coverage and high-frequency observations over China every 5 minutes. We assumed the land surface parameters are slow varying parameters, and meteorological data also have little different within 1 hour. Therefore, the FY-4A/AGRI China area observing data (5 minute), hourly meteorological data and monthly (16 Days) land surface parameters data were used to estimate PM2.5 and PM10 concentration every 5 minutes.
We have added the processing details in the revised manuscript. Please see Line 237-241 in Page 7 of the revised manuscript.
- P6L213-214 Please explain the strategy to divide the training set, testing set and validation set.
Reply: In this study, the integrated data set in 2021 were randomly and uniformly divided to traning set and testing set, and used in model training, testing, and model improvement. The validation set is the independent integrated data set in 2022, for data sample of 22980 points, which is used to evaluate the estimation model by comparing the concentrations of PM2.5 and PM10 estimation results and ground based sites measured results.
We have added the explanation of the strategy to divide the data set in the revised manuscript, please see Line 223-227 in Page 7 of the revised manuscript.
- P7L241-242 “The accuracy of the estimation results were tested by ground sites measurements”. If the ground site measurement is used as input of your model, it can not be used for validation!
Reply: The validation set is the integrated data set in 2022 (data sample of 22980 points), which is used to evaluate the estimation model by comparing the concentrations of PM2.5 and PM10 estimation results and ground based sites measured results. The ground site measurement in 2021 is used as input for model training and testing. Therefore, the validation set is completely independent data, and it has not used in developments of the model.
We have added the processing details in the revised manuscript, please see Line 261-264 in Page 8 of the revised manuscript.
- P7L246-247 The data sample for validation is 22980 points. According to your description, you use 2027 ground-based sites over the mainland of China with hourly data. And one year samples should be 17756520 and the matched data in 2022 is only 0.129%. There are only 10 samples per station on average. Please explain the reason for it. Please add a statistic for each station and the number of data samples in 2021 used for model training.
Reply: Thank you for carefully reviewing our manuscript. The validation data set were sampled from integration data in 2022, 12% of integration data in 2022 were randomly and uniformly sampled.
We have added the details of the validation data in the revised manuscript. Please see Line 261-264 in Page 8 of the revised manuscript.
Specific Comments
- P3L123-125 “CLM(CLound Mask product)product” product is redundant.
Reply: Thank you for helping us to check the typing mistake. We have revised in the manuscript. Please see Line 133 in Page 4 of the revised manuscript.
- P5l178 What is “SW”?
Reply: Thank you for carefully reviewing our manuscript. SW is an acronym which stands for short-wave. We have revised in the manuscript, please see Line 189 in Page 5 of the revised manuscript.
- Figure 4 The bin interval of the histogram in the figures is unclear.
Reply: We have redraw the difference histogram, the bin interval of the histogram is 1. Please see Figure 4 and Figure 6 in Page 8 and Page 9 of the revised manuscript.
- Figure 5-6 What’s the white color mean in the figures? Why are the regions with white color in (b) and (d) different? The value of 20 and 40 for PM2.5 and the value 50 and 100 for PM10 are difficult to distinguish in the figures.
Reply: We have redraw Figure 5-6, please see Figure 5-6 in Page 9 of the revised manuscript.
- P11L293-399 please explain why you choose the Shijiazhuang and Hotan station for the analysis.
Reply: The aerosols over Shijiazhuang are considered mainly sources of anthropogenic aerosols from biomass burning and industrial activities [5]. The Hotan city is surrounded by Taklimakan desert, the major aerosol over Hotan is dust aerosol during the dusty period [6]. The typical haze and dust storm occurred over northwest of China on March 10 and April 20, respectively. Therefore, we collected the air quality data measured by ground site (PM2.5 and PM10 concentrations) in Shijiazhuang (site number is 1029A, locate at 114.4422°E, 38.0444°N) and Hotan (site number is 3615A, locate at 79.9131°E, 37.113°N) during the haze and dust storm cases occurred on March 10 and April 20, 2022.
We have added the explanation in the revised manuscript, please see Line 356-365 in Page 14 of the revised manuscript.
- P13L342 ‘Altough’ to ‘Although’
Reply: Thank you for helping us to check the typing mistake. We have revised in the manuscript. Please see Line 403 in Page 16 of the revised manuscript.
References
- Chen, Z.-Y.; Zhang, T.-H.; Zhang, R.; Zhu, Z.-M.; Yang, J.; Chen, P.-Y.; Ou, C.-Q.; Guo, Y. Extreme gradient boosting model to estimate PM2.5 concentrations with missing-filled satellite data in China. Atmospheric Environment 2019, 202, 180-189, doi:https://doi.org/10.1016/j.atmosenv.2019.01.027.
- Gui, K.; Che, H.; Zeng, Z.; Wang, Y.; Zhai, S.; Wang, Z.; Luo, M.; Zhang, L.; Liao, T.; Zhao, H.; et al. Construction of a virtual PM2.5 observation network in China based on high-density surface meteorological observations using the Extreme Gradient Boosting model. Environment International 2020, 141, 105801, doi:https://doi.org/10.1016/j.envint.2020.105801.
- Zhou, X.; Cao, Z.; Ma, Y.; Wang, L.; Wu, R.; Wang, W. Concentrations, correlations and chemical species of PM2.5/PM10 based on published data in China: Potential implications for the revised particulate standard. Chemosphere 2016, 144, 518-526, doi:https://doi.org/10.1016/j.chemosphere.2015.09.003.
- Kong, L.; Xin, J.; Zhang, W.; Wang, Y. The empirical correlations between PM2.5, PM10 and AOD in the Beijing metropolitan region and the PM2.5, PM10 distributions retrieved by MODIS. Environmental Pollution 2016, 216, 350-360, doi:https://doi.org/10.1016/j.envpol.2016.05.085.
- Sun, X.; Yin, Y.; Sun, Y.; Sun, Y.; Liu, W.; Han, Y. Seasonal and vertical variations in aerosol distribution over Shijiazhuang, China. Atmospheric Environment 2013, 81, 245-252, doi:https://doi.org/10.1016/j.atmosenv.2013.08.009.
- Liu, H.; Wang, X.; Talifu, D.; Ding, X.; Abulizi, A.; Tursun, Y.; An, J.; Li, K.; Luo, P.; Xie, X. Distribution and sources of PM2.5-bound free silica in the atmosphere of hyper-arid regions in Hotan, North-West China. Science of The Total Environment 2022, 810, 152368, doi:https://doi.org/10.1016/j.scitotenv.2021.152368.

Reviewer 2 Report
This paper evaluated the applicability of FengYun satellites for inversion of atmospheric particulate matter concentration based on the machine learning model. The research methodology is basically reasonable, but the article lacks innovation, the methodology is not clearly presented, and the depth of content needs to be explored. In addition, there is much room for improvement in the article structure and language expression.
General comments:
In the third paragraph of the introduction, the authors mix together to introduce remote sensing spectral features (i.e., AOD and observed radiation) and inversion algorithms, which hardly makes sense logically. It has been said that AOD is superior in inversion of atmospheric particulate matter (Lines 49-64), but in the absence of any evidence, it is obviously hard to be convinced when it suddenly reaches lines 64-68 that AOD has a large error, and then says that the observed radiation is theoretically superior. In line 93 the authors mention that they have improved the XGBoost model, so what are the advantages of the XGBoost model over other algorithms, what are the shortcomings, and what improvements did the authors make? It is suggested that the authors split this section into two paragraphs to introduce the corresponding problems and research progress in terms of the spectral characteristics of atmospheric particulate matter and inversion algorithms, respectively, and to introduce the scientific issues of interest in this paper.
The method description is too general. It is suggested that the methods section details the XGBoost model architecture and what improvements the authors have made to the model. Also, how are the key model parameters determined and what are the exact values? How is the data of different resolutions interpolated and fused? How is the correlation between PM2.5 and PM10 considered and treated in the model?
The results are too superficial and the discussion too simplistic. I hope the authors will dig further in the depth of the result. For example: 1) What is the performance of the model in the test period and validation period respectively; 2) What is the spatial variation of the model accuracy and what are the influencing factors; 3) What are the temporal and spatial variation of the atmospheric particulate matter concentration based on satellite inversion and what are the influencing factors; 4) What are the differences in the characteristics of pm2.5 and pm10 at the model or mechanism level?
Finally, the language is too verbose. Many of the descriptions of the content of the figures are shown in the legend, and there is no need for textual descriptions. For example, Lines 145-147, 257-260, 333-335, 339-343 …..
Specific comments:
Line 123: The two products are duplicated.
Section 2.2: What is the temporal resolution of the data, what is the quality, are there any missing values, and how are they handled?
Line 228: Put the Table 2 into section 2, and add the temporal resolution of each variable.
Figure 3-6: Why are some of the frequencies in the figures fractional and some integers? Please add the horizontal and vertical axis units. PM2.5 is wrongly written in several graphs.
Figures 7-9: The concentration units are not consistent with the text.
Lines 299-301: “Some few regions” is grammatical error.
Lines 315: What is the haze recognition algorithm used here, please explain in the method.
Figure 10: Figure 10 can be merged with Figure 1 and is not necessary.
Lines 347-349: What are the times corresponding to the stages of formation, development and dissipation of particulate matter in the diagram, and what are the criteria for discriminating them?
Author Response
Responses to the comments from Reviewer2
Dear Reviewer,
Thank you for reviewing our manuscript. The valuable comments and suggestions have helped us to improve the paper. Please find below the comments in blue italics and our responses in black, the changes were highlighted in the revised manuscript.
Comments and Suggestions for Authors
This paper evaluated the applicability of FengYun satellites for inversion of atmospheric particulate matter concentration based on the machine learning model. The research methodology is basically reasonable, but the article lacks innovation, the methodology is not clearly presented, and the depth of content needs to be explored. In addition, there is much room for improvement in the article structure and language expression.
Reply: We thank the reviewer for the valuable comments and suggestions that have helped us to improve our manuscript. Following these comments and suggestions, we have taken a lot of efforts to make the article more concise and clear. The language editing services of MDPI have helped us to improve the language and readability of the manuscript.
General comments
- In the third paragraph of the introduction, the authors mix together to introduce remote sensing spectral features (i.e., AOD and observed radiation) and inversion algorithms, which hardly makes sense logically. It has been said that AOD is superior in inversion of atmospheric particulate matter (Lines 49-64), but in the absence of any evidence, it is obviously hard to be convinced when it suddenly reaches lines 64-68 that AOD has a large error, and then says that the observed radiation is theoretically superior. In line 93 the authors mention that they have improved the XGBoost model, so what are the advantages of the XGBoost model over other algorithms, what are the shortcomings, and what improvements did the authors make? It is suggested that the authors split this section into two paragraphs to introduce the corresponding problems and research progress in terms of the spectral characteristics of atmospheric particulate matter and inversion algorithms, respectively, and to introduce the scientific issues of interest in this paper.
Reply: Thank you for the valuable suggestion. We have revised this part of the manuscript following the suggestion. The section1 was split into two paragraphs and added citations to make the writing more concise and clear. Please see senction1 (Line 30-108 in Page 1-3 of the revised manuscript).
- The method description is too general. It is suggested that the methods section details the XGBoost model architecture and what improvements the authors have made to the model. Also, how are the key model parameters determined and what are the exact values? How is the data of different resolutions interpolated and fused? How is the correlation between PM2.5 and PM10 considered and treated in the model?
Reply: Thank you for the valuable suggestion.
The parameter tuning is essentially for improving the model and to achieve optimal performance. Therefore, we evaluated the model based on testing set, and parameter optimized in training. Then, the model parameters max_depth (Maximum depth of the tree) = 12, eta (learning rate) = 0.06, n_estimators (number of gradient boosting trees) = 160, subsample (Sampling rate of training samples) = 0.8, objective = reg:gamma were used in the study for PM2.5 and PM10 estimation. Moreover, the positive correlation between PM2.5 and PM10 is considered, and XGBoost model was rebuilt to an Multi-output model for estimating PM2.5 and PM10 concentrations in ground level synchronously.
We took different data matching methods in data integration and model application. In data integration, all kinds of data (FY-4A/AGRI data, Meteorological data, and Land surface parameters data) were sampled following ground sites location (nearest grid data from ground sites) to get hourly (temporal resolution of ground sites measurements) matched dataset. Then the matched data set was normalized, and the matched data set also divided into training set, testing set and validation set for model traning, testing and result validating respectively. In model application, both FY-4A/AGRI data, meteorological data, and land surface parameters data were projected and interpolated (using bilinear interpolation method) to 4 km resolution used for improved estimation model input, and the improved estimation model was applied to estimate PM2.5 and PM10 concentrations synchronously over mainland China every 5 minutes.
We have added the processing details in the revised manuscript. Please see Line 220-241 in Page 7 of the revised manuscript.
- The results are too superficial and the discussion too simplistic. I hope the authors will dig further in the depth of the result. For example: 1) What is the performance of the model in the test period and validation period respectively; 2) What is the spatial variation of the model accuracy and what are the influencing factors; 3) What are the temporal and spatial variation of the atmospheric particulate matter concentration based on satellite inversion and what are the influencing factors; 4) What are the differences in the characteristics of pm2.5 and pm10 at the model or mechanism level?
Reply: Thank you for the valuable suggestion. We have made further analysis of the result.
The Root Mean Squared Error (RMSE) is 2.56μg/m3 for PM2.5 concentrations, and 5.32 μg/m3 for PM10 concentrations in validation period; and 8.69 μg/m3 for PM2.5 concentrations, and 13.77 μg/m3 for PM10 concentrations in validation period.
The performance of the PM2.5 and PM10 concentration estimations were eval-uated in different months. The temporal evolution results indicates the characteristics of seasonal variation. Figure 8 shows the PM2.5 and PM10 estimation results have higher accuracy (lower RMSE) in warm seasons, and lower accuracy (higher RMSE) in cold seasons, this may result from the land cover changes. The land surface albedo is quite high and does not exhibit stable due to the snow cover in cold seasons, the relationships between visible and IR channels over bright regions resulting in large difficulties in land surface albedo estimation [1-6]. Therefore, the ground-level particulate matter concentration estimations may have larger uncertainties in cold seasons.
We have added the analysis in the revised manuscript. Please see Figure 8 and Line 312-322 in Page 11 of the revised manuscript.
- Finally, the language is too verbose. Many of the descriptions of the content of the figures are shown in the legend, and there is no need for textual descriptions. For example, Lines 145-147, 257-260, 333-335, 339-343 …..
Reply: Thank you for the valuable suggestion. We have revised the manuscript, and removed the verbose descriptions. Please see Line 157-158 in Page 5, Line 267-280 in Page 8, Line 285-299 in Page 9, and Line 356-365 in Page 14 of the revised manuscript.
Specific comments
- Line 123: The two products are duplicated.
Reply: Thank you for helping us to check out the typing mistake. We have revised in the manuscript. Please see Line 133 in Page 4 of the revised manuscript.
- Section 2.2: What is the temporal resolution of the data, what is the quality, are there any missing values, and how are they handled?
Reply: The China Environmental Monitoring Center (CNEMC) provide ground based sites measured concentrations of PM2.5 and PM10 data hourly. The ground based sites measured air quality data have been quality controlled following China's National Ambient Air Quality Standard (CNAAQS) to improve the data accuracy [7], the uncertainty of particulate matter concentration measurement is less than 5 μg/m3 [8]. In this study, we collected the sites measured concentrations of PM2.5 and PM10 from 2027 ground-based sites over mainland of China, the temporal ranges of the data are from 2021 to 2022. We conducted quality control on the hourly PM2.5 data to remove missing values and severe outliers of data in data integration.
We have added the processing details in the revised manuscript. Please see Line 150-153 in Page 4, and Line 157-158 in Page 5 of the revised manuscript.
- Line 228: Put the Table 2 into section 2, and add the temporal resolution of each variable.
Reply: Thank you for the suggestion. We have put the Table 2 into section 2, and add the temporal resolution of each variable. Please see Table 2 in Page 3 of the revised manuscript.
- Figure 3-6: Why are some of the frequencies in the figures fractional and some integers? Please add the horizontal and vertical axis units. PM2.5 is wrongly written in several graphs.
Reply: Thank you for helping us to check the typing mistake. The color of scatter plots represents the data frequency (fractional), and the y-axis of the difference histograms represents the number of bins (integers). We have revised and redrawn in the manuscript. Please see Figure 3-6 in Page 8-9 of the revised manuscript.
- Figures 7-9: The concentration units are not consistent with the text.
Reply: We have redraw Figures 7-9. Please see Figure 7 in Page 10, Figure 9 in Page 12, and Figure 10 in Page 13 of the revised manuscript.
- Lines 299-301: “Some few regions” is grammatical error.
Reply: Thank you for the helping us to check out the grammatical error. The language editing services of MDPI have help us to improve the language and readability of the manuscript. We have revised in the manuscript. Please see Line 309 in Page 10 of the revised manuscript.
- Lines 315: What is the haze recognition algorithm used here, please explain in the method.
Reply: The novel algorithm of haze identification was employed to detect haze regions [9]; the algorithm is built upon the spectral characteristics of different pixels: the visible band reflectance of the cloud pixels is greater than clear sky pixels, while the brightness temperature in the infrared channels is usually lower than the clear sky and haze pixels, and the brightness temperature difference is also higher than that of the clear sky and haze pixels. Then, the above characteristics were taken, and the adaptive improvements to the threshold selection were carried out for haze detection based on FY-4A/AGRI, and the PM2.5 and PM10 concentrations were estimated during the haze pollution processes.
- Figure 10: Figure 10 can be merged with Figure 1 and is not necessary.
Reply: Thank you for the suggestion. We have merged these two Figures. Please see Figure 1 in Page 5 of the revised manuscript.
- Lines 347-349: What are the times corresponding to the stages of formation, development and dissipation of particulate matter in the diagram, and what are the criteria for discriminating them?
Reply: Figure 11 shows the time series of particulate matter concentrations over ground based sites during haze and dust storm cases. The development process of ground level particulate matters during haze and dust storm cases were shown, but the formation and dissipation are hard to derive from the Figure. We have revised in the manuscript. Please see Line 373-375 in Page 15 of the revised manuscript.
References
- Hsu, N.C.; Tsay, S.-C.; King, M.D.; Herman, J.R. Aerosol Properties Over Bright-Reflecting Source Regions. IEEE Transactions on Geoscience and Remote Sensing 2004, 42, 557, doi:10.1109/tgrs.2004.824067.
- Sun, L.; Wei, J.; Bilal, M.; Tian, X.; Jia, C.; Guo, Y.; Mi, X. Aerosol Optical Depth Retrieval over Bright Areas Using Landsat 8 OLI Images. Remote Sensing 2016, 8, doi:10.3390/rs8010023.
- Román, M.O.; Schaaf, C.B.; Lewis, P.; Gao, F.; Anderson, G.P.; Privette, J.L.; Strahler, A.H.; Woodcock, C.E.; Barnsley, M. Assessing the coupling between surface albedo derived from MODIS and the fraction of diffuse skylight over spatially-characterized landscapes. Remote Sensing of Environment 2010, 114, 738-760, doi:https://doi.org/10.1016/j.rse.2009.11.014.
- Liu, J.; Schaaf, C.; Strahler, A.; Jiao, Z.; Shuai, Y.; Zhang, Q.; Roman, M.; Augustine, J.A.; Dutton, E.G. Validation of Moderate Resolution Imaging Spectroradiometer (MODIS) albedo retrieval algorithm: Dependence of albedo on solar zenith angle. Journal of Geophysical Research: Atmospheres 2009, 114, doi:10.1029/2008jd009969.
- Liang, S.; Fang, H.; Chen, M.; Shuey, C.J.; Walthall, C.; Daughtry, C.; Morisette, J.; Schaaf, C.; Strahler, A. Validating MODIS land surface reflectance and albedo products: methods and preliminary results. Remote Sensing of Environment 2002, 83, 149-162, doi:https://doi.org/10.1016/S0034-4257(02)00092-5.
- Jin, Y.; Schaaf, C.B.; Woodcock, C.E.; Gao, F.; Li, X.; Strahler, A.H.; Lucht, W.; Liang, S. Consistency of MODIS surface bidirectional reflectance distribution function and albedo retrievals: 2. Validation. Journal of Geophysical Research: Atmospheres 2003, 108, doi:10.1029/2002jd002804.
- Chen, Z.-Y.; Zhang, T.-H.; Zhang, R.; Zhu, Z.-M.; Yang, J.; Chen, P.-Y.; Ou, C.-Q.; Guo, Y. Extreme gradient boosting model to estimate PM2.5 concentrations with missing-filled satellite data in China. Atmospheric Environment 2019, 202, 180-189, doi:https://doi.org/10.1016/j.atmosenv.2019.01.027.
- Miao, Y.; Liu, S. Linkages between aerosol pollution and planetary boundary layer structure in China. Science of The Total Environment 2019, 650, 288-296, doi:https://doi.org/10.1016/j.scitotenv.2018.09.032.
- Si, Y.; Chen, L.; Zheng, Z.; Yang, L.; Wang, F.; Xu, N.; Zhang, X. A Novel Algorithm of Haze Identification Based on FY3D/MERSI-II Remote Sensing Data. Remote Sensing 2023, 15, doi:10.3390/rs15020438.

Round 2
Reviewer 1 Report
Authors have addressed most concerns in the previous review processes. The manuscript has a great improvement over the original submission. However, there are still some technical and language concerns in Revision. Authors should carefully consider all concerns below and revise the manuscript accordingly to make your manuscript easy to understand, at least no typos, no logic conflicts.
Section3.1 Step1 "The matched dataset in 2021 was randomly and uniformly divided into the training set and testing set" The percent of training set and testing set is also random?
Section3.1 Step3:According to step1 you use data at ground site locations for training and testing, the input data at step3 has a 4 km resolution. How do you realize the site prediction model to match the input data with 4km resolution? Especially the ground sites are unevenly distributed.
L228 "improvie" to "improve"
Author Response
Dear Reviewer,
Thank you for reviewing our manuscript. The valuable comments and suggestions have helped us to improve the paper. Please find below the comments in blue italics and our responses in black, the changes were highlighted in the revised manuscript.
Authors have addressed most concerns in the previous review processes. The manuscript has a great improvement over the original submission. However, there are still some technical and language concerns in Revision. Authors should carefully consider all concerns below and revise the manuscript accordingly to make your manuscript easy to understand, at least no typos, no logic conflicts.
Reply: We thank the reviewer for carefully reviewing our manuscript, the valuable comments that have helped us to improve our manuscript. Following these comments and suggestions, we have taken a lot of efforts to make the writing more concise and clear. We have carefully checked and revised the manuscript to avoid the ambiguity, typos and logic conflicts. The language editing services of MDPI also helped us to improve the language and readability of the manuscript.
1. Step1 "The matched dataset in 2021 was randomly and uniformly divided into the training set and testing set" The percent of training set and testing set is also random?
Reply: Thank you for the valuable comments. The percent of training set and testing set is not random, the matched dataset in 2021 was randomly and uniformly assigned, training set were composed by 80% of the matched dataset in 2021, and testing set were composed by 20% of the matched dataset in 2021. We have revised the description to avoid the ambiguity. Please see Line 227-229 in Page 7 of the revised manuscript.
2. Step3:According to step1 you use data at ground site locations for training and testing, the input data at step3 has a 4 km resolution. How do you realize the site prediction model to match the input data with 4km resolution? Especially the ground sites are unevenly distributed.
Reply: The Step3 is the model application, the improved model was used to estimate the concentrations of PM2.5 and PM10 synchronously every 5 minutes over mainland China, based on FY-4A/AGRI observed radiations, meteorological data and land surface parameters data. The input parameters are including FY-4A/AGRI observed radiations, meteorological and land surface parameters, and the model outputs are the concentrations of PM2.5 and PM10.
Therefore, the FY-4A/AGRI observed radiation data, meteorological data and land surface parameters data were each projected and interpolated (using the bilinear interpolation method) to a 4 km resolution used for the improved estimation model input to estimate concentrations of PM2.5 and PM10 in 4 km resolution. The FY-4A/AGRI observed radiation data, meteorological data and land surface parameters data are enough for model estimation, and it does not need to match the ground site data in the model application. The model training and validation need to match model input data (FY-4A/AGRI observed radiation data, meteorological data and land surface parameters data) with ground site measured concentrations of PM2.5 and PM10, the all kinds of data were sampled from ground site locations (the nearest grid data from ground sites) to match various datasets (Step 1).
3. L228 "improvie" to "improve".
Reply: Thank you for helping us to check the typing mistake. We have revised in the manuscript. Please see Line 230 in Page 7 of the revised manuscript.

Reviewer 2 Report
The authors may have misunderstood my suggestion. In my last general comments, I meant to split the third paragraph of the introduction to present issues and advances in the mechanism and methods of remote sensing inversion separately, rather than splitting the entire introduction section into two paragraphs.
The authors still do not provide a detailed description of the model used, and the few parameters listed are not known to have any specific meaning or effect. This is far from enabling the reader to understand the model, let alone reproduce their results.
Author Response
Dear Reviewer,
Thank you for reviewing our manuscript. The valuable comments and suggestions have helped us to improve the paper. Please find below the comments in blue italics and our responses in black, the changes were highlighted in the revised manuscript.
1. The authors may have misunderstood my suggestion. In my last general comments, I meant to split the third paragraph of the introduction to present issues and advances in the mechanism and methods of remote sensing inversion separately, rather than splitting the entire introduction section into two paragraphs.
Reply: Sorry for misunderstanding the suggestion. We have split the third paragraph of the introduction to present issues and advances in the mechanism and methods of remote sensing inversion separately. Please see Line 53-88 in Page 2 of the revised manuscript.
2. The authors still do not provide a detailed description of the model used, and the few parameters listed are not known to have any specific meaning or effect. This is far from enabling the reader to understand the model, let alone reproduce their results.
Reply: Thank you for the valuable suggestion. We have added the the specific meaning and effects of the model parameters. Moreover, the strategy of the model rebuilt details also added in the revised manuscript. We hope this would help readers to reproduce the results. Please see Line 233-251 in Page 7 of the revised manuscript.
